# Singular Adult Neural Stem Cells Do Not Exist

**DOI:** 10.3390/cells11040722

**Published:** 2022-02-18

**Authors:** David Petrik, Sara Jörgensen, Vasileios Eftychidis, Florian A. Siebzehnrubl

**Affiliations:** 1School of Biosciences, Cardiff University, Cardiff CF10 3AX, UK; jorgensens1@cardiff.ac.uk (S.J.); eftychidisv@cardiff.ac.uk (V.E.); 2School of Biosciences, European Cancer Stem Cell Research Institute, Cardiff University, Cardiff CF24 4HQ, UK

**Keywords:** neural stem cells, adult neurogenesis, cell lineage tracing, clonal analysis, cell heterogeneity, transcriptomics, self-renewal, neural progenitors

## Abstract

Adult neural stem cells (aNSCs) are the source for the continuous production of new neurons throughout life. This so-called adult neurogenesis has been extensively studied; the intermediate cellular stages are well documented. Recent discoveries have raised new controversies in the field, such as the notion that progenitor cells hold similar self-renewal potential as stem cells, or whether different types of aNSCs exist. Here, we discuss evidence for heterogeneity of aNSCs, including short-term and long-term self-renewing aNSCs, regional and temporal differences in aNSC function, and single cell transcriptomics. Reviewing various genetic mouse models used for targeting aNSCs and lineage tracing, we consider potential lineage relationships between Ascl1-, Gli1-, and Nestin-targeted aNSCs. We present a multidimensional model of adult neurogenesis that incorporates recent findings and conclude that stemness is a phenotype, a state of properties that can change with time, rather than a cell property, which is static and immutable. We argue that singular aNSCs do not exist.

## 1. Introduction

“Singularity is almost invariably a clue. The more featureless and commonplace a crime is, the more difficult it is to bring it home”. This is a quote from Sir Arthur Co-nan Doyle’s Sherlock Holmes story, “The Boscombe Valley Mystery”. The conundrum to solve here is the stemness of aNSCs. We wish that aNSCs possess unique, singular properties, which undoubtedly identify them. However, the latest results suggest that their features are more commonplace, and that it is difficult to define what an aNSC is.

## 2. Adut Neurogenic Niches

aNSCs exist in three discrete areas, so-called ‘niches’, of the adult mammalian brain: in the subgranular zone (SGZ) of the hippocampus, in the subventricular zone (SVZ) of the lateral ventricles, and in the walls of the 3rd ventricle (3V) surrounded by the hypothalamus [1,2,3,4], which we name here as the hypothalamic ventricular zone (HVZ). These niches are conserved in many mammalian species. The adult HVZ niche was observed in mice, rats, sheep [5], lemurs, and humans as containing aNSC marker-expressing ependymocytes called tanycytes [6,7,8]. Mice, rats, lemurs, and humans share a restricted co-expression of aNSC markers in the HVZ, but humans show an additional three aNSC populations [9]. In adult mice, lineage tracing studies confirm that tanycytes generate new neurons in the hypothalamus, including in the arcuate (ArcN) and ventromedial nuclei (VMN) [10,11,12]. In a comparative study of ovine, mouse, and human hypothalamus, Battalier et al., observed immature doublecortin (DCX)-positive and mature, Human Neuronal Protein C and D (HuC/D)-positive neurons occurring in hypothalamic nuclei [5]. This neurogenic potential of tanycytes in adult mice is further supported by their expression of the neurogenesis related protein doublecortin-like (DCL) [13], which was shown to co-localize with DCX in the mouse SVZ and SGZ [13,14]. In addition to rodents, SVZ and SGZ neurogenic niches have been confirmed in most mammalian species [15,16]. New neurons in the SGZ were confirmed by BrdU cell tracing in non-human primates [17] and by ^14^C dating in humans [18].

## 3. Neurogenic Rates in Different Niches

The rate of adult neurogenesis varies between the neurogenic niches and mammalian species. In the SGZ, neurogenesis accounts for an annual turnover rate of around 10–14% of hippocampal neurons in mice [18,19,20] and 1.75% in humans [18]. In contrast to the SGZ, although more than tens of thousands of neuroblasts migrate into the olfactory bulb from the SVZ every day, only a fraction survive to complete their differentiation [21]. Despite this low survival rate, all neurons in the deeper layers of mouse OB can be replaced in 12 months [22]. Adult neurogenesis in the HVZ is often reported as having much smaller cell turnover than both SGZ and SVZ, ranging between 1–37% [23]. However, this turnover can reach over 50% during the first 2.5 months of age in mice [24], which represents a significant neurogenic potential. It is important to consider that the rate of neurogenesis differs between species. New neurons daily produced in the SGZ count 9000 in rodents [25], 1300 in monkeys [26], and 700 in humans [18].

## 4. What Is a Stem Cell?

A stem cell has two key identifying properties. It is self-renewing and exhibits cellular potency for differentiated progenies. More elaborate definitions and stem cell properties have been formulated since the seminal discovery of the adult stem cells by Till and McCulloch [27,28]. However, given the latest research findings [29,30,31], we must ask if aNSCs fit this definition of self-renewing cells with cell lineage potency. In other words, is there a singular aNSC, or do we need to revisit our understanding of stemness in the adult brain?

As we discussed above, aNSCs from these niches differ in many of their properties including their cell dynamics, ontogeny, and origin. aNSCs in the SGZ (so called Type-1 cells) originate from neural progenitors developed (E16.5 in mice) from the dentate gyrus neuroepithelium [32]. During the early postnatal development, aNSCs are established from these dentate gyrus progenitors, which express *Nestin* or *Hopx* [33,34]. Similar to the SGZ, aNSCs of SVZ (so called B1 cells) are also generated from the embryonic neuroepithelium. Between E13.5 and E15.5 in mice, radial glia cells from the ventricular zone, which act as *Pax6*+ embryonic neural stem cells [35], give rise to aNSCs of the adult SVZ [36,37]. Finally, the origin of aNSCs (tanycytes) of the adult HVZ is traced to Sonic Hedgehog (*Shh*)-expressing progenitors from the embryonic floorplate [38,39].

Despite these ontogenic differences, all aNSCs can self-renew and possess the potency to generate one or more types of differentiated cell progeny. However, how many self-renewing cell divisions are necessary to satisfy the definition of self-renewal, especially if the cell’s life span is shorter than life span of the organism? Furthermore, would the ability of certain aNSCs to generate only one type of differentiated cells, so called ‘unipotency’, fulfil the rigorous definition of stem cells?

## 5. Stem Cell Traits in Progenitor Cells

The seminal paper by Encinas et al. [40] suggested that Nestin-expressing aNSCs in the SGZ only undergo three asymmetrical divisions to generate a progenitor cell. This implies that aNSCs, which do not last but terminally differentiate into an astrocyte, are, therefore, incapable of prolonged self-renewal. Such non-self-renewing aNSCs cannot be considered as stem cells. In contrast, another seminal study by Bonaguidi et al. [41] concluded that SGZ aNSCs also expressing Nestin are capable of asymmetric cell divisions and sustained self-renewal over almost the entire lifespan of the animal. While some of the differences between these two studies can be attributed to technical approaches (e.g., population vs. clonal analysis [42]), these studies revealed that not all aNSCs are the same in their self-renewal and potency. Indeed, hippocampal aNSCs expressing Nestin or the astrocyte-specific glutamate/aspartate transporter (GLAST) displayed different cell dynamics for generating adult-born neurons [43]. This suggests not only differential proliferation, quiescence, and self-renewal in the aNSC pool, but raises a question of stemness among aNSCs. To paraphrase—not all aNSCs uphold a rigorous definition of stem cells [44], something that is well-recognized in developmental biology, where “the origin of stem cell populations from progenitor cells does occur repeatedly in normal development with respect to the formation of the various types of tissue-specific stem cell” [45]. This apparent difference between aNSCs and their daughter progenitor cells is removed in a more encompassing term ‘neural stem and progenitor cells’ (NSPCs) used in both adult and embryonic neurogenesis [46,47]. Indeed, the ground-breaking work of Pilz et al. [29] showed that the so-called ‘transiently amplifying progenitors’ (TAPs or Type-2 cells) are capable of extended symmetric divisions, blurring the distinction between stem cells and progenitors. In addition, Pilz et al., observed that the so-called radial glia cell (RGC)-like aNSCs (Type-1 cells) sometimes directly differentiated into two neuronal cells. These two findings challenge not only the very definition of aNSCs and their stemness, but also the linear model of adult neurogenesis in SGZ [48].

## 6. A Limited Warranty of Stemness

A more recent study from the Jessberger lab [30] compared NSPCs using Ascl1 or Gli1 as genetic driver and demonstrated that these subpopulations differ in their self-renewal capacity; Ascl1-targeted cells were eventually lost after activation (consistent with the Encinas model), while Gli1-targeted cells showed long-term self-renewal and persistence (consistent with the Bonaguidi model). Together, these findings seem to indicate that there are long-term and short-term self-renewing aNSC populations in the hippocampus. This idea is supported by a recent report from the Bonaguidi lab comparing Nestin-targeted versus Ascl1-targeted aNSCs that demonstrated that Nestin-targeted aNSCs had capacity for long-term self-renewal while Ascl1-targeted aNSCs persisted only short-term [31]. It remains to be resolved whether Gli1-expressing aNSCs are giving rise to a short-term self-renewing, Ascl1-expressing stem cell subpopulation, or whether Gli1- and Ascl1-expressing aNSCs constitute separate entities. In addition, long term cell lineage tracing and clonal analysis experiments with the Gli1-CreER^T2^ line need to be performed to confirm if Gli1+ aNSCs display extended longevity. So far, lineage tracing with the Gli1-CreER^T2^ line was done for only 5 days [40], which is too short to address the longevity of Gli1+ aNSCs. It is further noteworthy that, so far, Aslc1-targeted, short-term aNSCs appear to generate only neuronal progenies, while Gli1- or Nestin-targeted long-term aNSCs make neurons and astrocytes.

Because of the stemness of neural progenitors, we can say that aNSCs possess limited warranty not only of their self-renewal but of their identity. We can also contemplate if what we call a neural progenitor may be a stem cell with temporarily limited or acquired self-renewal capacity. Thus, a stem cell is not a cell type but rather a phenotype, a state of properties that changes with time and interventions. However, if both aNSCs and progenitors can self-renew, which one is the stem cell, or are they both stem cells? Or, is only the original mother cell in the aNSC lineage the true stem cell, which we should define by another term, such as ‘root cell’?

## 7. Stemness as a Phenotype

The property of stemness is usually seen through the prism of cell heterogeneity. For example, *Ascl1* expressing aNSCs rapidly proliferate in the juvenile SGZ but increase their quiescence and self-renewal with age because of *Ascl1* depletion over time [31,49,50]. In contrast to *Ascl1*-positive aNSCs, *Nestin* or *Gli1* aNSCs in SGZ are more quiescent and last longer [30,31]. Reduction in Ascl1 expression over time, thus, promotes the stem cell phenotype, whereas Nestin or Gli1 aNSCs retain their stem cell phenotype and a higher quiescence over time. This perceived stem cell heterogeneity is a consequence of reductionistic approaches, which are necessary for technical and practical reasons. However, these apparently discrete aNSC subpopulations may be intermingled as we discuss in greater details below.

The field of multipotent hematopoietic stem cells (HSCs), which originated the stem cell field [49], offers an alternative perspective on self-renewal and stemness of aNSCs. There are dormant HSCs, which divide only five times over the lifetime of a mouse and can reversibly switch between active self-renewal and dormancy, maintaining a reservoir of most potent stem cells [50]. A deeply quiescent population of NSCs in the SVZ divide only every 3–5 months, comparable to the dormant HSCs [51]. Is it possible that some of the quiescent aNSCs in the hippocampus, such as the *Nestin* or *Gli1* expressing aNSCs [30,31], can repeatedly re-enter quiescence after a period of proliferation? Are these the ‘dormant’ aNSCs that maintain the so-called ‘neurogenic reserve’ [52]? Stemness may be adopted transiently as observed in neural progenitors [29] or elicited by injury even in cells that are not considered stem cells under physiological conditions, such as astrocytes (reviewed by [53]). However, this perspective of stemness as a phenotype and not a cell type brings us back to the opening question: how many (and for what period) self-renewing cell divisions are needed to define a cell as a stem cell?

## 8. Heterogeneity of Cell Potency

Unlike pluri-potent and multi-potent stem cells, including embryonic NSCs (reviewed in [47]), aNSCs potency or lineage differentiation is predominantly directed to specific neuron subtypes in vivo. This is particularly true in the SGZ, where aNSCs exclusively generate the granule cell neurons but not any other type of neurons, including dozens of types of interneurons of the dentate gyrus [54]. aNSCs in the SGZ possess an astrogliogenic capacity, which would make them bi-potent. However, as with self-renewal, it remains to be determined if this astrogliogenesis is confined to discrete aNSC subpopulations [31,41] or is the terminal, exhaustive differentiation step [40]. Using clonal analysis in GLAST-expressing aNSCs, we found a 2:1 ratio of active clones generating only neurons versus bi-potent clones [55]. When comparing terminally differentiated clones, this ratio changed to 10:1 in favour of neuron-only clones (unpublished observation), which may indicate that terminal differentiation into astrocytes is a rare event, at least in young adult mice. The frequency of neuron-only producing clones declines with age in favour of aNSC quiescence, while bi-potent clones seem to disappear completely in aged mice [31]. Because GLAST also targets parenchymal astrocytes, we could not quantify clones that may have terminally differentiated only into astrocytes. Interestingly, the recent in vivo imaging studies rarely observed astrogliogenesis in different hippocampal aNSCs subpopulations [29,30]. *Ascl1* expressing aNSCs generate only neurons, whereas *Gli1* expressing aNSCs can generate rare astrocytes [30]. By contrast, astrogliogenesis is a common feature of Nestin-targeted aNSCs [31]. This may imply the existence of different aNSC subtypes, with short-term aNSCs (*Ascl1*-positive) likely restricted to the neuronal lineage, while long-term aNSCs (*Gli1*- or *Nestin*-positive) generate both neurons and astrocytes.

In the SVZ and the HVZ, aNSC potency may be broader. Distinct, aNSCs in the SVZ (Type-B cells) generate a spectrum of olfactory interneurons [56] and possess an oligodendrogliogenic potency [57], a feature that is confined to a unique aNSC subpopulation [58]. In the HVZ, aNSCs can generate different types of neurons, such as proopiomelanocortin (POMC) or neuropeptide Y (NPY)-expressing neurons [12,59]. Given the potency diversity among aNSC subpopulations, even in a single niche, questions arise about how the potency defines their stemness. Is unipotency enough to define a stem cell, especially if it is short-lived? Are the unipotent, short-term aNSCs a subset, or even daughter cells, of bi-potent (or multipotent) long-term aNSCs? Or are there also different types of long-term aNSCs, each restricted to a single lineage (neuron or astrocyte)? Moreover, is the level of potency a phenotype, which is determined by time and interventions, as we suggested in the case of self-renewal? The very concept of aNSC potency also depends on how discriminately we define the differentiated cells in the cell lineage. Clearly, we can distinguish between astrocytes and neurons. However, can the capacity to generate distinct, yet ontogenetically or functionally similar neuronal subtypes be considered as multi-potency? These questions are not just an issue of semantics; they cause us to contemplate the very concepts of stem cell potency and stemness.

## 9. Stem Cell Heterogeneity from Single Cell Analyses

In the previous sections, we deliberated whether discrete subpopulations of aNSC exist in one or more neurogenic niches. Recent advances in single cell profiling provide some answers to this question (Table 1). Single cell transcriptomics has unveiled the diversity of aNSCs within the same niche, including aNSC subpopulations within the SVZ that are regionally separated and that produce different neuronal subtypes [60,61]. By contrast, while aNSCs in the SGZ constitute a heterogeneous cell population, no discrete subpopulations could be defined based on single cell profiling so far [62,63]. Instead, hippocampal aNSCs show a spectrum of gene expression profiles, which can be resolved into quiescent vs. activated or diving vs. non-diving states [30,62] but does not hint at discrete aNSC populations producing different progenies. This may be because the transcriptional profiles between different aNSC phenotypes are overlapping and, therefore, cannot be readily resolved without the use of specific Cre drivers to identify such aNSC states or phenotypes. Specific single cell analysis of Ascl1-targeted or Gli1-targeted aNSCs revealed that these two populations do not exhibit separable transcription profiles, even though small but important differences exist that may explain whether these cells return to quiescence or not [30]. Nevertheless, the largely overlapping transcriptional profiles of hippocampal aNSCs may indicate that stem cell phenotypes last temporarily instead of as discrete subpopulations of aNSCs. Whether there exist regional subtypes of aNSCs in the hippocampus (e.g., ventral vs. dorsal) that produce different progenies (e.g., neuronal vs. astroglial) remains elusive.

## 10. Stem Cell Identity from Single Cell Analyses

In several single cell profiling studies, transcriptomes from quiescent aNSCs and astrocytes show a high degree of overlap, reinforcing the close relationship between the two cell types and the notion of stem cells as specialised astrocytes [62,63]. A more recent single cell analysis of embryonic and early adult hippocampal cell types could separate aNSCs and astrocytes and their developmental trajectories [64]. A different single cell study identified five distinct astrocyte subtypes with a unique subset that could be linked to neurogenesis and another potential intermediate progenitor population [65]. Direct comparison of aNSCs across the three neurogenic niches using single cell OMICs may help determine the relationship and heterogeneity between SVZ, SGZ, and HVZ aNSCs. Cross-comparisons of stem cells at different developmental time points (embryonal, early postnatal, and adult) have highlighted that aNSCs are distinguishable from embryonic or juvenile stem cells, thus demonstrating the importance of developmental context beyond postnatal stages for adult neurogenesis [34,64].

Nevertheless, several limitations of single cell transcriptomics need to be considered. Currently, the read depth of single cell transcriptomics is much shallower compared to conventional RNA sequencing, which limits the ability to identify subpopulations based on rarer transcripts. The ability to discriminate rare subpopulations, such as heterogeneous aNSCs, further depends on the number of cells that are profiled. Earlier studies typically profiled fewer cells in total and were therefore less likely to identify smaller subpopulations. Additionally, apparent subpopulations on single cell cluster analyses may reflect a phenotypic diversity, rather than genuine subpopulations.

## 11. Regional Heterogeneity of Stem Cells

In addition to aNSC heterogeneity, regional organisation of aNSCs generating specific progenies was found in the SVZ, which affects production of neuronal subtypes as well as oligodendrocytes [60,62,63]. Single cell profiling studies show regionally different transcriptomes in aNSCs within the SVZ, but it remains unclear whether aNSC heterogeneity is influenced by regionally different microenvironments or by developmental programs [69]. Transient ablation of proliferating NSCs in the brain using chemical or radiation damage has demonstrated that the niche is repopulated from persisting, quiescent NSCs once the ablation is stopped [43,51,70,71,72,73]. However, such paradigms cannot evaluate the ability of aNSCs to repopulate different neurogenic niches. A combination of single cell profiling and transplantation studies may resolve this question eventually. When aNSCs from the SVZ were transplanted into the hippocampus, they failed to generate granule cell neurons [74], but would aNSCs from the anterior SVZ be able to integrate into the posterior SVZ and generate local cell types and vice versa? In the HSC field, different stem cell populations can be functionally discriminated by their differential ability to repopulate the myeloablated bone marrow niche and their capacity for repeated transplantation. These paradigms have led to the identification and experimental validation of clearly defined subpopulations of stem and progenitor cells (e.g., long-term vs. short-term self-renewing stem cells, multi-lineage and lineage-restricted progenitors). Prospective isolation and transplantation of HSCs has been refined down to the level of single cell transplants that could repopulate the niche [75]. Single cell analyses have documented heterogeneity of HSCs and progenitor cells [76] and it will be interesting to see how this heterogeneity relates to bone marrow reconstitution studies. Paradigms to discriminate different subtypes of stem or progenitor cells are still lacking in the aNSC field. Such paradigms would not necessarily need to rely on transplantation to repopulate the niche, but they would need to allow to discriminate between long-term and short-term renewing cells. Combining such approaches in animal models with different aNSC reporters (e.g., Nestin-Cre, Ascl1-Cre, Gli1-Cre, GLAST-Cre) and long-term analysis of clonal progenies may demonstrate the capacity of these putative aNSC subtypes for long-term self-renewal and repopulation (Table 2). Recent studies indicate that Ascl1-targeted aNSCs constitute short-term self-renewing stem cells in the SGZ [30,31], but whether similar aNSC types exist in other neurogenic niches remains unclear. Multiple consecutive rounds of ablation and repopulation could be used to rigorously test the self-renewing capacity of the repopulating stem cell population. This may improve our ability to distinguish between stem and progenitor cells and shed light on putative heterogeneity of aNSCs in different neurogenic regions.

## 12. Heterogeneity of Fate Choice in Stem Cell Progenies

Indeed, the relationship between Nestin-, Ascl1-, Gli1- and GLAST-targeted aNSCs remains to be determined. Lineage tracing of Nestin-CreER^T2^ aNSCs show a slower, steady and non-exhaustive neurogenesis suggestive of long-term cell maintenance [43,81], something confirmed by clonal analyses studies that also detected extended quiescence with increased number of cell divisions [31,41]. On the other hand, GLAST-CreER^T2^ aNSCs are initially very active [82], generating new neurons in a steady (in SGZ) or additive (in SVZ) manner [19]. However, their neurogenesis is exhaustive [43], with increasing asymmetric cell divisions and depletion of their numbers over time in both neurogenic niches [77,78]. While very informative, these studies have utilized individual driver lines (Table 3) without testing if there is a change in expression of key stemness genes (such as Ascl1) in Nestin- or GLAST-CreER^T2^ labelled cells over time, for example. In other words, there may be aNSCs that express Ascl1 initially in young animals, which promotes activation and aNSC depletion; yet, these same aNSCs may downregulate Ascl1 and upregulate Nestin or Gli1 expression with increasing age to promote extended maintenance (as described in [82]).

In vitro, aNSCs generate neurons, astrocytes, and oligodendrocytes [90]; demonstrating this tri-lineage differentiation is one of the standard paradigms of testing stemness in the CNS. This further suggests that potency and stemness of aNSCs are enforced by the niche and may be unlocked as in the case of reactive astrocytes [91]. In vivo, the differentiation of aNSCs progenies is more limited, often to a specific subset of neurons (see above). On one hand, this shows that aNSCs possess the general potency to produce neurons and macroglia; on the other hand, it implies that regional heterogeneity of aNSCs is driven by environmental factors, rather than cell-intrinsic programs. However, one must bear in mind that the process of extracting aNSCs for in vitro experiments is de facto an injury, which may reprogram aNSCs into a more potent state. Reactive astrocytes become more plastic when isolated and cultured in vitro [91], which may be a specific property of reactive astrocytes generated by aNSCs in response to injury [92] and similar mechanisms may act directly on aNSCs as well. Nevertheless, in vitro studies were instrumental in establishing the stem cell nature of aNSCs, particularly because demonstrating multipotency is more challenging in vivo. Similarly, extended self-renewal capacity over time is easier to demonstrate in vitro, e.g., through serial passaging.

## 13. Temporal Heterogeneity of Stem Cells

Another unresolved question is whether aNSCs phenotypes change over time. Single cell profiling highlighted largely similar transcriptional profiles of aNSCs in the hippocampus, with discernible features related to quiescent or activated states. Ascl1-targeted aNSCs in SGZ are more likely to revert to a quiescent state (reflective of a long-term self-renewing aNSC) with increasing age [82]. Thus, are these aged aNSCs still the same cell as their younger, short-term self-renewing counterparts? Would their transcriptional profiles more resemble those of long-term self-renewing aNSCs? If so, this would indicate that stem cell phenotypes can be temporarily adopted and that there is a degree of plasticity between these phenotypes, which would also fit with their largely overlapping gene expression profiles (i.e., changing only a small number of transcripts can convert short-term self-renewing to long-term self-renewing stem cells). Parabiosis studies suggest that circulating factors affect neurogenesis levels in aged animals [93], indicating that complex interactions between aNSCs and their niche may influence stem cell phenotypes, and that age-related changes may be reversible by altering the niche. In support, circulating cytokines in the blood of younger mice increased neurogenesis levels in aged mice through remodelling of the niche [94]. It remains to be shown whether systemic factors directly act on aNSCs, and/or whether ‘rejuvenation’ of adult neurogenesis shifts quiescence vs. activation levels of aNSCs similar to those in younger animals. In support, blocking ageing-related tyrosine kinase signalling partially reverted aNSC quiescence [31]. We have argued that stemness is a phenotype that changes with time and interventions. Thus, environmental factors (e.g., cytokines from the blood, or age-associated changes of the brain milieu) will affect the number and type of self-renewing divisions that aNSCs undergo, influencing temporary stemness phenotypes that may be observable in younger, but not in aged, animals. Finally, ageing affects changes in aNSC differentiation patterns (e.g., reduced neurogenesis), but the underlying mechanisms remain largely unclear. A recent study demonstrated that loss of post-translational glycosylation of STAT3 results in age-associated changes of hippocampal aNSCs differentiation [95]. In addition to such cell-intrinsic mechanisms, environmental cues may also affect the altered fate of aNSC progenies in the aged brain.

## 14. Technical Influences on Stem Cell Heterogeneity

Heterogeneity of aNSCs may result from reductionistic models depending on different genetic drivers of reporter expression (or other genetic manipulation), such as the Ascl1-, GLAST-, Gli1-, and Nestin-Cre models. Partial overlap between genetic drivers in aNSCs and progenitor cells may influence the observed outcomes of stem cell divisions (discussed above). For instance, *Ascl1* is expressed in both long-term and short-term self-renewing aNSCs, with low *Ascl1* expression in Type-1 cells of the SGZ or B cells of the SVZ and high Ascl1 levels in corresponding intermediate progenitor cells [83]. A seminal study directly compared Ascl1- and Gli1-targeted progenies in the hippocampus by live-cell imaging and found that only Gli1-targeted aNSCs showed long-term self-renewal [30]. As discussed above, it remains to be determined whether *Ascl1* and *Gli1* target different aNSC subpopulations (long-term vs. short-term self-renewing), or whether *Gli1* is hierarchically superior to *Ascl1*. Nevertheless, these observations indicate the heterogeneity of stem cell subpopulations. We have used the GLAST-CreER^T2^ model [96] to trace clonal dynamics of hippocampal aNSCs and found that asymmetric divisions of aNSCs are linked to self-renewal, whereas symmetric divisions resulted in aNSC depletion [55]. We also observed clonal heterogeneity, with most clones persisting over the 4-week chase period, but approximately 1 in 5 clones terminally differentiated, comparable to a previous study [41].

## 15. Regionality of Stem Cell Heterogeneity

Ventricular neurogenic regions seem to possess more complex aNSC heterogeneity than the hippocampus. There are regionally distinct (i.e., dorsal vs. ventral) subpopulations of aNSCs in the SVZ [61]. In the adult medial-basal hypothalamus (MBH), expansion of the cell lineages occurs at the level of parenchymal progenitors [11], as is the case in the SVZ [77]. However, it remains to be determined if this is a specific cell dynamic in the cell lineage from *Fgf10*-expressing aNSCs [11], or the parenchymal expansion is a general feature in the MBH, as it is suggested to be in the *Sox2*-expressing lineage [97]. The fact that adult MBH contains aNSC in both mice and humans [9] and generates new orexigenic and anorexigenic neurons has been established beyond a reasonable doubt (reviewed in [3]). However, the cell identity, stemness, and lineage progression from aNSCs to neurons in MBH is less clear. Using the GLAST-CreER^T2^ line [19,43], Robins et al. showed that GLAST-expressing tanycytes, so-called α2, that line the dorso-medial part of 3V appear first after genetic recombination that labels them and their cell progeny, suggesting that these are the aNSCs of the HVZ [12]. On the other hand, the ventral, so called β tanycytes of the medial eminence, appear later, suggesting that they are derived from α2 tanycytes and may be the neural progenitors [12]. However, do β tanycytes serve as transient progenitors or rather active aNSCs in the sense of their self-renewal and stemness? Shorter time-points after Tamoxifen induction and transgenic mouse lines specific only to tanycytes should address this question. Indeed, the unique Rax-CreER^T2^ line expressed in tanycytes and not in astrocytes [98] confirms that α2 tanycytes generate β tanycytes [59]. However, this study also demonstrated the expansion of tanycyte-derived cells in the parenchyma, where DCX+ progenitors and neurons are not confined to the ventricular niche as in the SVZ [2], but dispersed in the MBH of both mice and humans [5]. Clearly, additional cell lineage tracing and clonal analysis studies are needed to determine the cell stages of the neurogenic process in the HVZ. These studies should address if there are specific aNSC subpopulations dedicated to generating discrete neuronal subtypes, as is the case in the SVZ. Finally, a combination of long-term time-lapse imaging and new, cutting-edge cell tracing technologies, such as iCOUNT [99], will address the stemness of progenitors and elucidate the relationship between cell heterogeneity and stemness.

## 16. Multidimensional Model of Neurogenesis

Results from studies on cell heterogeneity, stemness, and stem cell maintenance suggest two interpretations. First, there are discrete subpopulations of aNSCs, which generate independent clonal lineages of cell lines. Second, these apparently discrete subpopulations are a result of reductionistic technical approaches, but, in fact, they can change over time and with external influences such as disease. In other words, it remains to be tested whether some *Ascl1*-positive aNSCs downregulate *Ascl1* and upregulate *Gli1* expression over time, for example. We incline towards the second possibility, based on the observation that biological phenomena occur in gradients [100].

The linear model of adult neurogenesis assumes a one-way generation of differentiated cell progeny [48]. The conveyor belt starts from the RGC-like aNSCs and through transiently amplifying progenitors ends with adult-generated neurons. However, we discussed that at least some neural progenitors possess stem cell properties. In addition, two aspects of their identity remain to be directly observed. First, are there any progenitors that persist in the brain for the same duration as RGC-like aNSCs, ideally over the majority of the organism’s lifespan? Second, are progenitors able to generate RGC-like aNSCs? In vivo time-lapse imaging showed neural progenitors lasting and self-renewing for several weeks [29,30]. However, to determine if progenitors can persist as long as quiescent aNSCs, a longer lasting in vivo time-lapse imaging would need to be combined with a clonal analysis based on a gene, which is expressed exclusively in progenitors but not in aNSCs. Unfortunately, SGZ progenitors (Type-2a cells) share cell markers with aNSCs (Type-1 cells) [101]. The best option for a progenitor-specific cell marker may be Tbr2, which is expressed in neural progenitors. A transgenic inducible Tbr2-2A-CreER^T2^ line is available [102], however, to our knowledge, has not been used for lineage tracing in adult neurogenesis.

The prospect of progenitors generating stem cells seems unlikely at first glance. This is because our established perception of aNSCs is heavily dependent on morphological criteria, which dictate that aNSCs should have RGC-like morphology [103]. In addition, RGC-like aNSCs remain the main surviving cell stage after ablation of neurogenesis [43,69,71], suggesting that they are the stem cells. Therefore, it seems absurd, almost sacrilegious to suggest that progenitors not only act as NSCs but generate RGC-like aNSCs. However, the ablation experiments only show that a subset of RGC-like aNSCs are quiescent, whereas mitotically active aNSCs and progenitors are ablated. The ablation results do not rule out that certain progenitors have extended self-renewal. Moreover, stem cells can be generated by progenitors if we define them by their developmental status and potency [45]. To demonstrate whether progenitor cells can regenerate quiescent aNSCs, the opposite experiment would need to be performed—exclusive ablation of the non-proliferating stem cells. The recent in vivo time-lapse imaging studies [29,30], while ground-breaking and technically marvelous, take pictures once per day, an imaging frequency too low for directly observing a possible generation of RGC-like aNSCs from progenitors without the radial glia morphology. The concept that neural progenitors that do not exhibit radial glia morphology can act as stem cells is not impossible. For example, changes in *Ascl1* expression over the lifespan may determine the exhaustive mode of adult neurogenesis under the assumption that *Ascl1* is expressed in RGC-like aNSCs [82]. However, *Ascl1* expression is stronger in progenitors than in aNSCs [83], raising the possibility that *Ascl1* could label late-stage aNSCs that are on the cusp of turning into progenitors and that have already lost the radial glia morphology. This would be an alternative explanation of the limited self-renewal of *Ascl1*-targeted cells. This possibility that progenitors without RGC-like morphology can serve as stem cells has been clearly demonstrated during neuronal embryonal development. In developing neocortex, basal progenitors display stem cell behaviours while lacking the apical-basal connections and the standard RGC-like morphology [104].

Now, let’s assume two premises. First, there is a gradient of stemness and stem cell identity in adult NSCs. Second, lineage progression may be a two-way street, at least during the initial stages. Under these two premises, we propose an alternative model of adult neurogenesis, which considers different dimensions including time, stemness and cell identity (Figure 1). In this model, aNSCs may change their identity, stemness and marker expression over time or under influence of external interventions (e.g., disease, running) and neural progenitors may serve as stem cells and may generate aNSCs. The first premise (gradients) implies a spectrum of stemness that ranges from quiescent aNSCs to self-renewing progenitor cells [29,30]. We may distinguish long-term and short-term self-renewing cells along this spectrum. Another implication of a stemness gradient is that there is a stochastic probability for the outcome of any stem cell division. The most likely outcome is a division that maintains the aNSC and generates a progenitor that is designated to differentiate, which may be influenced by the division plane [55]. However, occasionally, a stem cell may directly differentiate into two neurons, or a progenitor cell may reconvert into a stem cell. We observed a low percentage of clones containing only progenitors 1 day after recombination (unpublished observation), which significantly increased with ablation of the stem cell transcription factor *Zeb1* [55]. While this indicates that aNSCs occasionally undergo symmetric division to produce two progenitor cells, it does not preclude the possibility that one of these progenitors may retain a stem cell phenotype. The stochastic probability of self-renewal effectively self-limits the renewal of short-term stem or progenitor cells. If a progenitor holds a 50% chance of undergoing self-renewal with each division, the overall chance of successive self-renewing divisions decreases exponentially because the progenitor is lost as soon as it does not undergo self-renewal. The probability for each division outcome is affected by cell type, age, location of the cell, and external influences.

In conclusion, aNSCs and neural progenitors reside along a spectrum of stemness, which affects their ability to self-renew over extended periods. Likewise, regionally heterogeneous aNSCs align along a spectrum of gene expression profiles that are similar but not identical. Therefore, a singular adult neural stem cell does not exist.

## Figures and Tables

**Figure 1 cells-11-00722-f001:**
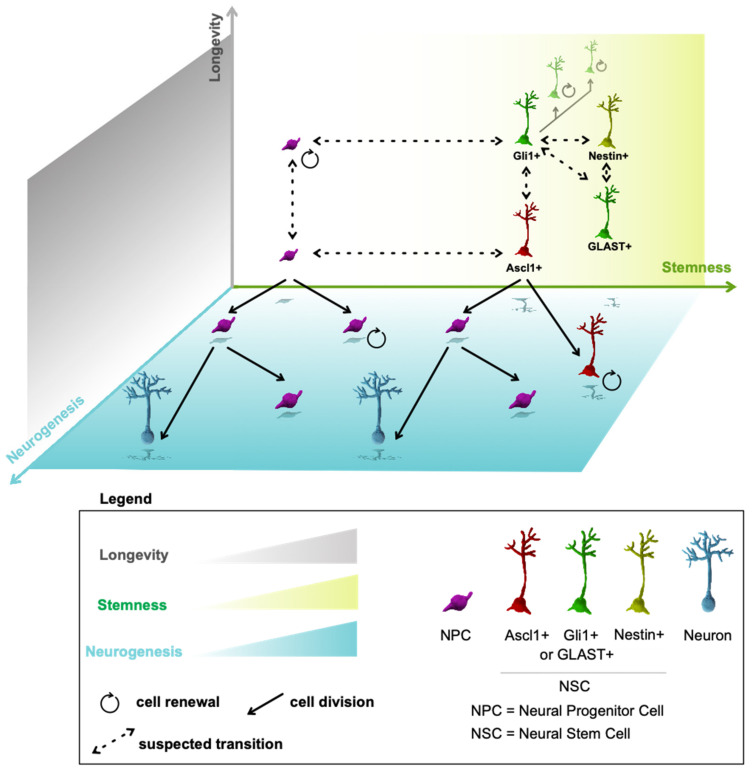
A multidimensional model of adult neurogenesis. Neurogenesis is conceptually asserted by three criteria: stemness, longevity of cells, and the neurogenesis process. On the plane of stemness as a function of longevity, self-renewing, long-term NSCs (such as Gli+ or Nestin+ NSCs) that last in the neurogenic niche longer are situated in the right top corner. Short-term NSCs that support exhaustive neurogenesis (such as Ascl1+ NSCs) are situated in the right bottom corner. GLAST+ NSCs reside between these two types on the stemness plane. Differentiating cell progeny of NSCs progresses on the right side of the neurogenesis plane. NPCs may acquire extended self-renewal and stem-like properties and last as long as some NSCs (in the left top corner). If NPCs self-renew only for few cell divisions (in the bottom left corner), they eventually transform into differentiating cell progeny along the horizontal plane of neurogenesis (on the left side). This process corresponds to the conveyor belt model of neurogenesis. It remains to be determined if there are long-term NPCs that can self-renew for longer periods of time. The long-term (Gli1+ and Nestin+) NSCs are also able to generate NPCs, however, this is not depicted to unclutter the diagram. It remains to be determined if long-term NSCs defined by a certain cell marker (i.e., Nestin) are just a segment of another long-term NSC pool (labelled by Gli1, for example) or these are truly discrete cell populations. Similarly, it remains to be determined if long-term NSCs can change their phenotype to short-term NSCs (i.e., Ascl1+ or GLAST+) and vice versa.

**Table 1 cells-11-00722-t001:** Overview of single cell transcriptomics studies with a focus on aNSCs outside of a disease or transgenic model. DG: dentate gyrus; SVZ: subventricular zone; OB: olfactory bulb; TF: transcription factor.

Reference	Region	Isolation and Sequencing	aNSC Hetero-Geneity	Notes
[63]	DG	Microdissection; negative selection (GluR1-, Cd24-); SORT-seq	quiescent v activated	Populations of quiescent and activated NSCs could be defined, but no other heterogeneity
[30]	DG	Intravital imaging, Microdissection, Gli1/Ascl1-CreER^T2^; TdTomato, Smart-seq2	quiescent v activated	Transcriptional differences partly overlapping amongst two subpopulations of NSCs (quiescent/activated) with a differential self-renewal capacity.
[62]	DG	Microdissection, Nestin-CFP^nuc^, SMART-seq	quiescent v activated	Identified two super-groups with six subgroups of NSC immediate progeny, corresponding to quiescent/activated states.
[64]	DG from embryonic and adult between E16.5 to P132	Microdissection, positive selection (hGFAP-GFP+), Fluidigm C1, 10X Chromium V1/2, Illumina HiSeq2000/2500/4000	developmental; lineage; young v aged	Single cell profiling of cell types in DG across prenatal, juvenile and adult. Neuronal intermediate progenitors (nIPCs), neuroblasts and immature granule cells did not form separate clusters in the transition from perinatal to adult, but radial glia molecularly switch at P5
[65]	DG	Whole hippocampus dissection, positive selection (ACSA-2+), modified SMART-Seq2, Illumina NextSeq 500	regional	Astrocyte clustering into 5 subgroups reveals intra- and inter-regional heterogeneity. Two distinct clusters are defined, one cluster spatially mapped to most GLAST+ cells in SGZ and an intermediate GLAST+ progenitor population mapped in subpial, stratum lacunosum moleculare, and DG
[66]	SVZ	Microdissection, positive and negative selection (Gfap-Gfp+, Prom1+, Egfr+/−, Cd31−, Cd24-, Cd45−); Fluidigm C1 Single-Cell Auto Prep chip and SMARTer-seq	quiescent v activated	Three NSC subpopulations across a spectrum of activation/differentiation states. Identified rare intermediate states with unique molecular fingerprints.
[67]	SVZ	Microdissection, Nestin-CreER^T2^ Histone H2B-Gfp, Diphtheria toxin receptor, positive and negative selection Glast+, Cd133+, Cd45− Cd95	quiescent v activated; young v aged	Analysis of NSCs from infancy to old age to identifies transition from quiescence to proliferation and uncovers NSC heterogeneity.
[61]	SVZ	Microdissection; hGFAP-GFP; 10X Chromium V3	regional	Two populations of NSPCs in dorsal vs. ventral V-SVZ are transcriptionally distinguishable.
[60]	SVZ Lateral v Septal walls	Microdissection, hGFAP::CreER^T2^; R26R^CAG-tdTomato^, Microwell and DROP-seq	regional; male v female	Regional and sex differences between lateral and septal wall NSCs. Distinct spatiotemporal TF expression profiles of dormancy and lineage progression across neurogenesis and oligodendrogenesis.
[68]	SVZ OB	Microdissection GFAP-CreER^T2^ Nestin-FlpER, Microwell/DROP-seq and SCOPE-seq	lineage	Heterogeneous qNSCs with distinct OB interneuron and astrocyte lineages. Identified novel V-SVZ proliferation marker in a transitory intermediate NSC population.
[69]	Embryonic cortex from 4 developmental timepoints between E11.5 to E17.5	Microdissection, CD1 mice, DROP-seq, and FISH and immunostaining Of adult V-SVZ	developmental time	Identification of embryonic cortical radial precursors with distinct transcriptional identity which is maintained through their transition to quiescence. A distinct E17.5 radial precursor population transcriptionally similar to adult V-SVZ qNSCs.

**Table 2 cells-11-00722-t002:** Mouse models used for clonal analysis. HVZ (hypothalamic ventricular zone), SVZ (subventricular zone), SGZ (subgranular zone), NSC (neural stem cell), d (day), mo (months), w (week), dpi (days post induction), mpi (months post induction), n.d. (not determined), quiescent NPs (quiescent neural progenitors), DG (dentate gyrus) GCL (granular cell layer), OB (olfactory bulb), * (additional comments can be found in Notes), ↑↓ (alternated), ↑ (increased), ↓ (decreased), ↔ (maintained/no change).

Driver	Reference	Mouse Line	Region	Quiescence	Active	Exhaustive (Short Term Maintenance)	Maintenance (Long Term Maintenance)	Self-Renewal	Notes
Nestin	[41]	Nestin-CreER^T2^: Z/EG	SGZ	↑↓	↓↑	↓ Assumed–activated RGLs maintained at 12 months	↑ (12 mo) *	↑	Reporter-positive radial glia-like cells displayed both self-renewal properties and multipotent differentiation at 2 mpi. Radial glial like cells can alternate between an active and quiescent state. * Maintenance of some activated radial glia-like cells up to 1 year.
		Nestin-CreER^T2^: MADM		↑↓	↓↑	↓ Assumed—activated RGLs maintained at 12 months	↑ (12 mo)	↑	Frequencies of all types of clones (quiescent, symmetrically self-renewed, asymmetrically self-renewed, and differentiated) were comparable between the Z/EG and MADM reporters. However, the MADM reporter allowed for a more rigorous clonal analysis of quiescent radial glia-like cells.
	[31]	Nestin-CreER^T2^: Confetti	SGZ	↑ (4 mpi in 6 mo old mice) “by calculating the time to cell-cycle entry and re-entry according to power-law decay fitting of clonal tracings”	↑ (Slow)	↓	↑ (4 mpi in 12 mo old mice)	↑	Nestin-NSCs are longer lived and slowly generate new neurons, astrocytes and NSCs. Nestin-NSCs prolong their quiescence with each division and switch to symmetric cell fate choice after NSC homeostasis has been lost in mice around 4–6 mo of age.
		Ascl1-CreER^T2^		n.d.	↑ (Fast)	↑	↓ (6 mo)	↓ *	Ascl1-NSCs demonstrated short term stem cell maintenance for approximately 1 week followed by rapid initial depletion that slowed with time. * No significant expansion (symmetric self-renewal) over time was observed in the Ascl1- NSC population.
GLAST	[77]	GLAST- CreER^T2^: Confetti	SVZ	↓	↑	↑	↓ (4–6 mo)	n.d. Suggested limited self-renewal	The NSC population underwent multiple rounds of division in a short time span, generating progeny before becoming exhausted. While other previously quiescent NSCs becomes activated to counteract the decline in adult neurogenesis.
	[78]	GLAST-CreER^T2^: Confetti	SVZ	n.d. Not determined in this mouse model	↑	↑	↓ (56 d)	n.d. Not determined in this mouse model	By 21 dpi, most clones consisted of progenitor cells or progenitor cells and neurons. By 56 dpi, the proportion of clones comprised by only neurons had increased. These clones were rarely found in the same hemisphere as a radial astrocyte, indicating NSC exhaustion to be the major terminating mechanism of OB neurogenesis.
	[55]	GLAST-CreER^T2^	SGZ	↔ *	↓ **	↑	↓ (4 w)	↓	* Depletion of Zeb1 does not directly alter the quiescent population. ** Active clones containing radial glia-like cell and non-radial glia-like cell progenies were significantly reduced, while depleted clones containing only differentiated progeny were significantly increased.
Troy	[79]	TroyGFP^iresCreER^	SVZ	↔ *	↑↓ Active NSCs can return to quiescence after one or more rounds of cell division	↓	↔ (32 w)	↑ **	From 14 dpi, and in subsequent timed points, both the density of NSC retaining clones and their stem cell content remained stable. * Most clones consisted of a single qNSC through all time points. ** At early time points, clones consisted of multiple Troy+ cells. Suggesting symmetric division upon activation.
		Ki67^iresCreER^	SVZ	↔	↑↓ *	↓	↑ (1 y)	↑	* Majority of active NSCs exit the cell cycle quickly, however some expand before returning to quiescence (qNSCs). These qNSCs may remain long-term to later contribute to ongoing adult neurogenesis.
VCAM1	[80]	VP lentivirus injection in Ai14 mice	SGZ	↔	↑ (Slow)	↓	↔ (28 d) *	↓	* Reporter-positive cells exhibited slow proliferation with some VCAM1-expressing NSCs remaining quiescent.
Hopx	[34]	Hopx-CreER^T2^	SGZ	↑	↑ *	↓ *	↑ (12 mo)	↑	Reporter-positive radial glia-like cells were quiescent neural progenitors with some capacity to self-renew. Notably, these qNSCs retain the capacity to re-enter the cell cycle up to a year post induction. * At 4 mpi, there was a large shift toward clones consisting of only mature neurons, indicating that some radial glia-like cells were depleted.

**Table 3 cells-11-00722-t003:** Overview of different models used for lineage tracing across different neurogenic niches, * (additional comments can be found in Notes), ↑↓ (alternated), ↑ (increased), ↓ (decreased), ↔ (maintained/no change).

Driver	Reference	Mouse Line	Region	Quiescence	Active	Exhaustive (Short Term Maintenance)	Maintenance (Long Term Maintenance)	Self-Renewal	Notes
Nestin	[81]	Nestin-CreER^T2^	SGZ	n.d.	↑	↓ *	↔ (100 d)	n.d.	Stem-like recombined cells with radial glia morphology was present in the SGZ up to 100 dpi * 50% of YFP+ cells expressed NeuN by 65 d and plateaued over subsequent time points.
	[40]	Nestin-CreER	SGZ	↔	↑	↓ *	↔ (45 d)	n.d.	Production of mature astrocytes detected after 20 d. * The fraction of labelled quiescent NPs, new astrocytes, and newly generated neurons remained constant over all time points (45 d).
GLAST	[19]	GLAST-CreER^T2^	SGZ	n.d.	↑ (4 mo)	↓ *	↔ (9 mo) *	n.d.	Reporter-positive mature neurons reached a plateau after 4 mo in the DG (also observed in the GCL of the OB). * Reporter-positive slow-dividing stem cells remained stable over months.
			SVZ	n.d.	↑	n.d.	n.d.	n.d.	The proportion of neurons in the GL of the OB increases linearly due to the net addition of inhibitory interneurons.
	[82]	GLAST- CreER^T2^	SGZ	n.d. Quiescence established not in this mouse line	↑ 1 to 3 self—renewing div.	↓ * Assumed non exhaustive as 28% of NSCs self-renew	↑ (30 d) **	↑	* 28% of stem cells underwent 3 or more self-renewing divisions before losing their stem cell identity in adults compared to 12% in juvenile mice. ** Increased self-renewal in adult mice is a mechanism contributing to preserving the NSC pool.
		Ki67-CreER^T2^		↑ (5 d)	n.d.	n.d.	n.d.	n.d.	In 1-month old mice NSCs remain proliferating (Ki67+). In contrast, in 6-month-old animals a significant proportion of NSCs returns to quiescence (Ki67-).
	[43]	GLAST-CreER^T2^	SGZ	n.d.	↑	↑	↓ (180 d)	n.d.	At 180 d post induction (dpi), many reporter-positive cells matured into neurons with a corresponding decrease in proportion of radial glia-like cells.
		Nestin-CreER^T2^		n.d.	↑	↓	↔ (180 d)	n.d.	There was an initial surge of reporter-positive cells through 30 dpi, which was followed by a plateau at later time points. Most reporter-positive cells were early progenitors at 12–60 dpi. At 180 dpi, cells were almost exclusively neurons or radial glia-like cells.
	[55]	GLAST- CreER^T2^	SGZ	↓	↑	↑	↓ (12 w)	↓	Steady decline of activated radial glia-like cells lead to the continuous recruitment of quiescent radial glia-like cells. In turn, resulting in exhaustion of the cell pool.
NG2	[12]	NG2-CreER	HVZ	n.d.	↑	↓	↔ (60 d)	↑ *	The absolute number of reporter positive NG2 glia remained constant up to 60 dpi. However, by day 60 the proportion of oligodendrocytes increased while the NG2 positive glia decreased. * The absolute number of NG2 glia remained constant between 7 d-60 dpi, indicating that the rate of cell death or differentiation was roughly the same, as they were generated by self-renewing divisions.
Fgf10	[11]	Fgf10—CreER^T2^	HVZ	n.d.	n.d.	↑ Number of Xgal^+^ tanycyes drops in adult	↓ (83 d) Number of Xgal^+^ tanycytes drops in adult	n.d.	The total number of reporter-positive cells found in adult mice showed a small but nonsignificant drop at 39–83 d compared to 24–27 d.
Ascl1	[29]	Ascl1-tdTomato	SGZ	↓	↑	↑	↓ (2 mo)	↓ “self-renewal capacity of Ascl1-targeted R cells is temporally limited”	By implanting a cortical window that allowed for 2-photon imaging, it was shown that, once activated, Ascl1-targeted radial glia-like progenitor cells generateA a burst of neurogenic activity to then commit to differentiation and loss. These cells did not re-enter long term quiescence.
	[83]	Ascl1-CreER^T2^	SGZ	↔ * No change in Sox2+ cells	↓ By 180 d	↓ Number of Sox2+ cells (180 vs. 30 d) is maintained and NeuN+ cells increase	↔ (180 d) *	↑	At 180 dpi, 65% of reporter-positive cells were NeuN positive granule neurons. However, 25% of reporter-positive cells also expressed markers of progenitor cells. *No obvious loss of Sox2+ cells indicate labeling of quiescent Type-1 cells.
			SVZ	n.d.	↑	↓ Sox2+ cells remained in the SVZ at 180 d	↑ (180 d) *	↑	At 30 dpi, many reporter-positive cells in the OB co-expressed NeuN, demonstrating that labelled cells are migrating and maturing. * Reporter-positive cells still remained in the SVZ and expressed Sox2, DCX or Ki67 up to 180 d after induction.
	[84]	Ascl1-CreER^TM^	SGZ	n.d.	↑	↑	↓ (180 d)	↓	30 d after induction, 86% of reporter-positive cells were mature neurons. This increased to 98% after 6 mo. Ascl1+ cells were mostly identified as Type 2a progenitor cells, but also a subset of stem cells with limited self-renewal potential.
Sox	[85]	*Sox2-CreER*	SVZ, SGZ	n.d.	↑	↓	↑ (4 mo)	↑	Reporter-positive cells with morphological characteristics of radial glia stem cells remained abundant in both brain regions up to 4 mo after induction.
	[86]	*Sox1*-tTA; LC-1; R26eYFP	SGZ	↓↑ *	↑↓ * Some NSCs diff., some remain NSCs	↓ Some Sox1+ remain NSCs	↑ (18 w)	n.d.	* A continuous, long term (3 mo) production of progenitors and NBs from Sox1+ cells is consistent with a stem cell population with long term neurogenic potential that alternate between an activated and a quiescent state. However, the decline of Sox1+ radial astrocytes after a 12-week chase period indicates that some cells permanently exit the stem cell pool.
Hopx	[87]	Hopx^CreER/+^	SGZ	n.d. Not determined in this mouse model	↑	↓ *	↑ (2 mo) *	n.d.	At 2 mpi, many reporter-positive cells differentiate into granule neurons and the proportion of NSCs declined. * At 2 mpi Sox+ and GFAP+ NSCs derived from Hopx+ NSCs were still identifiable.
Hes5	[88]	Hes5-CreER^T2^	SGZ	n.d.	↑	↓ *	↔ (100 d) *	n.d.	29% of reporter-positive NSCs remained 100 d post induction, with a corresponding increase in proportion of neuroblasts and postmitotic neurons. * The number of NSCs remained constant over 100 d post induction.
	[89]	Hes5-CreER^T2^	SVZ	n.d. Not determined in this mouse model	↑	↓ *	↑ (100 d)	n.d. Not determined in this mouse model	Reporter-positive cells in the SVZ continued to generate mitotic progenitors and neuroblasts 100 d after induction. * The neural stem cell population remained in the niche over months and retained long term neurogenic potential.
Troy	[79]	TroyGFP^iresCreER^	SVZ	n.d.	↑	↓	↑ (1 y)	n.d.	Reporter-positive cells remained in the SVZ up to 1 y post labelling while generating new neuroblasts.
PDGFRb	[58]	PDGFRb-P2A-CreER^T2^	SVZ	n.d. Not determined under physiological conditions	↑	↓	↑ (4 mo)	n.d.	Reporter-positive radial cells (GFP+GFAP+), TAPs, and migrating neuroblasts could be found within the SVZ at both 30 and 120 dpi, indicating that reporter-positive stem cells in the SVZ generate progeny up to 4 mpi.
VCAM1	[80]	Ai14 Cre (VP lentivirus injection)	SGZ	↔ *	↑ Analysed only at 28 dpi	↓ Quiescent NSCs remain constant	↔ (28 d) * Quiescent NSCs remain constant	n.d.	At 28 dpi 31% of reporter-positive cells were co -labelled with S100β+ and 67% were NeuN positive. * The ratio of quiescent NSCs that display a radial and horizontal morphology remained constant from 14 dpi to 28 dpi.
Spot14	[46]	Spot14-CreER^T2^	SGZ	n.d. Not determined in this mouse model	↑	↑	↓ (3 mo)*	n.d. Not determined in this mouse model	At 3 mpi, 62% of reporter-positive cells were mature neurons compared to 0% at 10 dpi. * The proportion of radial NSPCs declined from 48% to 24% (10 d vs. 3 mo) and non-radial NSPCs declined from 50% to 8%.

## Data Availability

Not applicable.

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
