# Peer review of "Singular Adult Neural Stem Cells Do Not Exist"

_cells, 2022, doi:10.3390/cells11040722_

Round 1

Reviewer 1 Report

Petrik et al present a compelling and thoughtful review of the state of understanding in definitions of adult neural stem cells in mammalian brain. Overall, I found the review engaging and thought-provoking. My comments focus on a few papers that are not discussed but I think should be and some subtopics that warrant more discussion.

In the opening paragraph a more thorough definition of what the authors mean by cell phenotype versus a singular cell type would help to frame the argument presented in this manuscript and help readers less familiar with the field better understand the nuances of the discussion within.

There are several papers that are not discussed currently but seem like they warrant inclusion. Hochgerner et al 2018, Nature Neuroscience used single cell RNAseq to profile numerous cell types in the DG from embryonic ages to adult but does not seem to be mentioned. Importantly, they were able to distinguish astrocytes from RGLs, a point relevant to discussion around line 180. Much of the work in Hochgerner does focus on juvenile and embryonic NSCs, which are not the focus of this review but they included adult NSCs too and therefore seem like a necessary inclusion in the discussions of single cell data relevant to defining aNSCs, as well as in the summary table.

Battiuk et al., 2020, Nature Communications similarly has single cell data from adult DG that seems relevant to this review. Particularly relevant to the points about transcriptional separation of astrocytes from RGLs,  Battiuk et al set out to characterize astrocytes in the hippocampus and found among their astrocytes an RGL population. This paper also seems worthy of inclusion in the discussions about single cell data and in the summary table.

More elaboration about how the 3 major aNSC niches compare would be helpful, especially for readers less familiar with the field. Specifically, a few lines describing conservation of these 3 niches across species would help: which species are known to have each niche? Which species might have all 3? There are a lot of unknowns here because, outside of mice and rats, investigation is limited, but some hints of the overlaps and the gaps in knowledge would be useful. Also, some statements about the relative quantities of aNSCs and their neurogenic rates between niches would be useful. In lab mice, the SVZ is far more neurogenic than the DG which is far more neurogenic than the hypothalamus. It would improve the readers global understanding of these processes to get some idea of that comparison. It also helps to frame why so little data is available on the hypothalamus compared to the other niches.

The authors do an excellent job of comparing driver models used to identify and trace aNSCs. Given how much data rely on these drivers, some information about the functions of the proteins driven by these promoters endogenously do would be helpful. For example, Nestin is an intermediate filament protein. If the authors have any speculation about why these different drivers might mark different states of aNSCs (for example, something related to the function of the endogenous protein), that would also be a nice addition. Also, could the authors give an idea of how much overlap might be expected between drivers if they could be combined. For example, if Nestin drivers label 75% of RGL cells (as defined by something the GFAP/Sox2+morphology) and Gli1 drivers label 75% of that same GFAP/SOX2-defined population, these drivers obviously must overlap somewhat. This would be a helpful consideration to introduce to the reader if there are known data on it. Such overlap could feed in to discussion of the authors’ proposed definition of aNSCs by phenotype which gives them a probability bias for different subsequent division/fate decisions, rather than them being defined as completely separable populations.

In the section about using single cell data to find subpopulations, some cautionary discussion about interpretation of single cell data would help frame things better. Specifically, the ability of single cell to differentiate subpopulations depends on cell input numbers—the more cells and the higher their quality, the easier it is to pick up more subpopulations. So, in past studies with hundreds of cells, perhaps aNSCs show up as one cluster but if you could input 1000 cells, maybe more subdivisions would become evident. In a related note, seeing separate clusters in single cell does not imply separate cell populations, but can in fact just be phenotype. For example, with enough cells from a cycling population, you can get separate clusters for different phases of the cell cycle. These seem like relevant points to consider when evaluating single cell data for the purposes of determining what kind of heterogeneity is evident in aNSCs.

The authors focus on adult NSCs but it would helpful context to mention a bit about the developmental trajectory of NSCs through postnatal development. Hochgerner et al and Berg et al 2019, Cell Stem Cell provide helpful comparisons showing substantial postnatal refinement in NSC properties, making adult NSCs unique from juvenile and embryonic counterparts. Some brief treatment of this idea would help readers less familiar with the field who may be tempted to think of anything postnatal as ‘adult.’

In the discussion about in vitro NSCs: do in vitro a NSCs have similar potential when derived from each of the 3 niches? The present discussion talks about in vitro aNSCs as one class. Is that the case? Also, additional discussion of how in vitro studies have been used to supplement in vivo findings that establish aNSCs as stem cells could be added to the section discussing the pitfalls of in vitro methodologies to provide a frame of reference for this discussion.

Is there a reason GLAST+ NSCs are not included in Fig 1? I really like the figure and the other major driver lines all seem to be represented but not GLAST.

The addition of a few summary paragraphs/sentences to some subsections may be useful to establish how the presented findings provide evidence to support the claim that stemness is a phenotype. For instance, discussion of how stem cell heterogeneity lends to stemness as a phenotype after the heterogeneity of cell potency and/or single cell analyses subsections. A similar statement/paragraph could also be added after the first paragraph in the heterogeneity of fate choice subsection, summarizing how the relationship between the different aNSC driver lines and stemness genes supports stemness as a fluid phenotype.  A final summary statement/paragraph could be added to the temporal heterogeneity subsection summarizing how temporal changes to quiescent and activation states of aNSC supports the argument to classify stemness as a phenotype.  

Author Response

Petrik et al present a compelling and thoughtful review of the state of understanding in definitions of adult neural stem cells in mammalian brain. Overall, I found the review engaging and thought-provoking. My comments focus on a few papers that are not discussed but I think should be and some subtopics that warrant more discussion.

We thank the reviewer for their kind comments.

In the opening paragraph a more thorough definition of what the authors mean by cell phenotype versus a singular cell type would help to frame the argument presented in this manuscript and help readers less familiar with the field better understand the nuances of the discussion within.

We thank the reviewer for this suggestion. In the abstract, we added an explanation of what is meant by “cell phenotype” versus a “cell type” in lines 15-17. This explanation is further expanded in the original submitted text of last paragraph of section “Stem cell traits in progenitor cells” (lines 129-136) and in the section “Stemness as a phenotype” (lines 137-161).

There are several papers that are not discussed currently but seem like they warrant inclusion. Hochgerner et al 2018, Nature Neuroscience used single cell RNAseq to profile numerous cell types in the DG from embryonic ages to adult but does not seem to be mentioned. Importantly, they were able to distinguish astrocytes from RGLs, a point relevant to discussion around line 180. Much of the work in Hochgerner does focus on juvenile and embryonic NSCs, which are not the focus of this review but they included adult NSCs too and therefore seem like a necessary inclusion in the discussions of single cell data relevant to defining aNSCs, as well as in the summary table.

This reference is now included in the text and table 1. We have updated our discussion in lines 228-239 of the revised manuscript. In addition to the scRNAseq discussion, we included a new Table 2 on clonal analyses of aNSCs.

Battiuk et al., 2020, Nature Communications similarly has single cell data from adult DG that seems relevant to this review. Particularly relevant to the points about transcriptional separation of astrocytes from RGLs, Battiuk et al set out to characterize astrocytes in the hippocampus and found among their astrocytes an RGL population. This paper also seems worthy of inclusion in the discussions about single cell data and in the summary table.

We have included this reference in the text and table 1. Discussion of this reference is in lines 228-232 in the revised manuscript.

More elaboration about how the 3 major aNSC niches compare would be helpful, especially for readers less familiar with the field. Specifically, a few lines describing conservation of these 3 niches across species would help: which species are known to have each niche? Which species might have all 3? There are a lot of unknowns here because, outside of mice and rats, investigation is limited, but some hints of the overlaps and the gaps in knowledge would be useful. Also, some statements about the relative quantities of aNSCs and their neurogenic rates between niches would be useful. In lab mice, the SVZ is far more neurogenic than the DG which is far more neurogenic than the hypothalamus. It would improve the readers global understanding of these processes to get some idea of that comparison. It also helps to frame why so little data is available on the hypothalamus compared to the other niches.

We have added two new paragraphs at the introduction of the review (lines 28-59). The first paragraph (“Adult neurogenesis in three neurogenic niches”) compares the 3 major aNSC niches, emphasizing differences between different species. The second paragraph (“Neurogenic rates in different niches”) discusses rates of neurogenesis in the 3 major aNSC niches, which reflect the abundance of aNSCs.

The authors do an excellent job of comparing driver models used to identify and trace aNSCs. Given how much data rely on these drivers, some information about the functions of the proteins driven by these promoters endogenously do would be helpful. For example, Nestin is an intermediate filament protein. If the authors have any speculation about why these different drivers might mark different states of aNSCs (for example, something related to the function of the endogenous protein), that would also be a nice addition. Also, could the authors give an idea of how much overlap might be expected between drivers if they could be combined. For example, if Nestin drivers label 75% of RGL cells (as defined by something the GFAP/Sox2+morphology) and Gli1 drivers label 75% of that same GFAP/SOX2-defined population, these drivers obviously must overlap somewhat. This would be a helpful consideration to introduce to the reader if there are known data on it. Such overlap could feed in to discussion of the authors’ proposed definition of aNSCs by phenotype which gives them a probability bias for different subsequent division/fate decisions, rather than them being defined as completely separable populations.

We thank the reviewer for these suggestions. The information about the different mouse lines using the different promoters can be found in Table 3 (former Table 2). However, the molecular mechanisms that may cause different cell dynamics are poorly understood. Therefore, a discussion why aNSCs labelled by Nestin drivers differ from those labelled by GLAST drivers could only be highly speculative. Furthermore, this is beyond the focus of this review, which is the identity of aNSCs. Nevertheless, we have added a couple of sentences on the longevity of Gli1+ aNSCs in lines 124-128.

In the section about using single cell data to find subpopulations, some cautionary discussion about interpretation of single cell data would help frame things better. Specifically, the ability of single cell to differentiate subpopulations depends on cell input numbers—the more cells and the higher their quality, the easier it is to pick up more subpopulations. So, in past studies with hundreds of cells, perhaps aNSCs show up as one cluster but if you could input 1000 cells, maybe more subdivisions would become evident. In a related note, seeing separate clusters in single cell does not imply separate cell populations, but can in fact just be phenotype. For example, with enough cells from a cycling population, you can get separate clusters for different phases of the cell cycle. These seem like relevant points to consider when evaluating single cell data for the purposes of determining what kind of heterogeneity is evident in aNSCs.

This is an excellent suggestion with which we fully agree. We have included a discussion on the limitations of single cell studies in lines 235-247.

The authors focus on adult NSCs but it would helpful context to mention a bit about the developmental trajectory of NSCs through postnatal development. Hochgerner et al and Berg et al 2019, Cell Stem Cell provide helpful comparisons showing substantial postnatal refinement in NSC properties, making adult NSCs unique from juvenile and embryonic counterparts. Some brief treatment of this idea would help readers less familiar with the field who may be tempted to think of anything postnatal as ‘adult.’

While we agree with the reviewer that the developmental trajectories of aNSCs are an interesting topic, the space needed for an informative discussion would go beyond the scope of our review. We have included the Hochgerner study in the discussion (lines 228-239) and table 1 and we hope this will point the interested reader in the right direction. We did not include the Berg et al study in Table 1, because it did not use single cell transcriptomics, which is a clear focus of that table. Nevertheless, we have included a clarification of the period of adult neurogenesis that includes both references (lines 234-239).

In the discussion about in vitro NSCs: do in vitro a NSCs have similar potential when derived from each of the 3 niches? The present discussion talks about in vitro aNSCs as one class. Is that the case? Also, additional discussion of how in vitro studies have been used to supplement in vivo findings that establish aNSCs as stem cells could be added to the section discussing the pitfalls of in vitro methodologies to provide a frame of reference for this discussion.

The potential of NSCs in vitro depends partly on the culture conditions. Many studies have tried to optimise conditions that favour the differentiation into specific cell types, but discussing this literature is beyond the scope of our review. Our point is that in vitro conditions are likely reprogramming aNSCs into a more primitive ‘ground state’. The absence of tissue specific cues makes it difficult to compare the potential of aNSCs from different niches in an in vitro environment. We have broadened discussion of in vitro studies as the reviewer suggested (lines 314-317).

Is there a reason GLAST+ NSCs are not included in Fig 1? I really like the figure and the other major driver lines all seem to be represented but not GLAST.

We thank the reviewer for spotting this oversight and have amended Fig. 1 which now also includes GLAST+ cells.

The addition of a few summary paragraphs/sentences to some subsections may be useful to establish how the presented findings provide evidence to support the claim that stemness is a phenotype. For instance, discussion of how stem cell heterogeneity lends to stemness as a phenotype after the heterogeneity of cell potency and/or single cell analyses subsections. A similar statement/paragraph could also be added after the first paragraph in the heterogeneity of fate choice subsection, summarizing how the relationship between the different aNSC driver lines and stemness genes supports stemness as a fluid phenotype. A final summary statement/paragraph could be added to the temporal heterogeneity subsection summarizing how temporal changes to quiescent and activation states of aNSC supports the argument to classify stemness as a phenotype.

We thank the reviewer for the suggestion of additional subheadings. We added new subheadings in lines 28, 48, 112, 225, 335, and 374 to separate specific sub-topics and to improve the structure of the text.

Reviewer 2 Report

This review is complex and challenging. It puts together well-organized data from the recent single cell transcriptomics and from the cell tracking studies in adult mouse brain, looking to the three neurogenic niches. The authors provide one table with the transcriptomic studies and one table with the tracking studies, together with a short description of each study, followed by pertinent comments. However, are these data really reflected in the proposed “unified” multidimensional model of the adult neurogenesis or is this model misleading? The authors should clarify some statements and speculate less, in order this work to bring a significant contribution and to have a relevant meaning.

Major concerns

A distinction of different types of tissue stem cells is necessary. Their heterogeneity in each “class” /tissue is supported by many recent data, including some in this review. The authors mention the hematopoietic stem cells (HSCs) as an adult stem cell prototype, with the potential to differentiate into all blood lineages. Many recent studies including single cell transcriptomics revealed heterogeneity in HSCs, both in adult and during development. So, “singular adult HSCs do not exist ”, maybe this assessment is valid only in the early embryo.

 By sure the brain is more complex and heterogeneous than other tissues, and its adult stem cells are by sure heterogeneous. We do not expect singular adult neural stem cells to exist! At least not in the adult brain, maybe only the early neural plate!

On this line, the reviewed studies tried to define this heterogeneity, but also to classify the cells according to their similitudes and differences, isn`t it? This would be a very good goal for classifying data reviewed here, coming from the proliferating cells (long or short term), from the quiescent and from the differentiated cells in each of the three neurogenic niches.

 Another confusion is related to the definition of a neural stem cell, here and elsewhere. The “old” definition of a neural stem cell is that it self-renews and has the potential to differentiate into neurons, astrocytes and oligodendrocytes. Not all at the same time, but each in different conditions. In the neurogenic niches, the neurogenic cells can be stem cells or progenitor cells. The presented tracking studies tried to clarify this problem. What are the arguments for changing this definition?

 Anyway, the situation of the neural stem cells is different in any of the three neurogenic niches, and it is related to their different ontogeny. The proposed model interprets data from the SGZ.  The situation in the VZ-related niches (SVZ and HVZ) is more complex. It would be interesting to present them separately and to take the conclusion afterwards. Otherwise, the presented conclusions are too speculative. Some assertions are not necessary, such as the transplantation of OB neurons in hippocampus. However, I would recommend clarifying the development origin of each of the three niches, for the reader to understand the differences between their stemness, composition and cell fates, as coming from the reviewed studies here.

Author Response

This review is complex and challenging. It puts together well-organized data from the recent single cell transcriptomics and from the cell tracking studies in adult mouse brain, looking to the three neurogenic niches. The authors provide one table with the transcriptomic studies and one table with the tracking studies, together with a short description of each study, followed by pertinent comments. However, are these data really reflected in the proposed “unified” multidimensional model of the adult neurogenesis or is this model misleading? The authors should clarify some statements and speculate less, in order this work to bring a significant contribution and to have a relevant meaning.

We thank the reviewer for their critical comments, which have helped identify areas in our manuscript that could be improved for clarity. The model we propose is adding to previous concepts in the field as it incorporates information on different subtypes of aNSCs, which are the basis of the transcriptomic and tracking studies that are presented in the tables. We hope the reviewer will agree that the revised manuscript is clearer and more to the point.

Major concerns

A distinction of different types of tissue stem cells is necessary. Their heterogeneity in each “class” /tissue is supported by many recent data, including some in this review. The authors mention the hematopoietic stem cells (HSCs) as an adult stem cell prototype, with the potential to differentiate into all blood lineages. Many recent studies including single cell transcriptomics revealed heterogeneity in HSCs, both in adult and during development. So, “singular adult HSCs do not exist ”, maybe this assessment is valid only in the early embryo.

The reviewer raises an important point: stem cell heterogeneity permeates all systems. We have amended our discussion of HSCs in lines 271-273 to reflect HSC heterogeneity as evidenced from single cell transcriptomics studies.

By sure the brain is more complex and heterogeneous than other tissues, and its adult stem cells are by sure heterogeneous. We do not expect singular adult neural stem cells to exist! At least not in the adult brain, maybe only the early neural plate!

We are happy that the reviewer agrees with us on this important point. We think that this point bears repeating even though it may be obvious to the reviewer. We hope our review will contribute to spreading this important message in the field of neurogenesis and beyond.

On this line, the reviewed studies tried to define this heterogeneity, but also to classify the cells according to their similitudes and differences, isn`t it? This would be a very good goal for classifying data reviewed here, coming from the proliferating cells (long or short term), from the quiescent and from the differentiated cells in each of the three neurogenic niches.

Comparing proliferating cells originating from quiescent or more differentiated cells is an interesting suggestion. However, it goes beyond the scope of this review and there are not enough details from the primary research articles for such comparison at the present time. Nevertheless, we included a new Table 2 on clonal analyses of aNSCs, which improves the resources and discussion on cell heterogeneity and cell dynamics.

Another confusion is related to the definition of a neural stem cell, here and elsewhere. The “old” definition of a neural stem cell is that it self-renews and has the potential to differentiate into neurons, astrocytes and oligodendrocytes. Not all at the same time, but each in different conditions. In the neurogenic niches, the neurogenic cells can be stem cells or progenitor cells. The presented tracking studies tried to clarify this problem. What are the arguments for changing this definition?

We are not sure if we follow the reviewer on this point. Our manuscript clearly defines (neural) stem cells as self-renewing and having potential to differentiate. We do not wish to change that definition. But it is important to highlight areas where NSCs are stretching the limits of this definition, some of which are demonstrated by recent studies that are discussed in our review.

Anyway, the situation of the neural stem cells is different in any of the three neurogenic niches, and it is related to their different ontogeny. The proposed model interprets data from the SGZ.  The situation in the VZ-related niches (SVZ and HVZ) is more complex. It would be interesting to present them separately and to take the conclusion afterwards. Otherwise, the presented conclusions are too speculative. Some assertions are not necessary, such as the transplantation of OB neurons in hippocampus. However, I would recommend clarifying the development origin of each of the three niches, for the reader to understand the differences between their stemness, composition and cell fates, as coming from the reviewed studies here.

We have introduced a new paragraph on the developmental origin of adult NSCs in the section “What is a stem cell?” (lines 70-78).

While we agree that it would be interesting to compare different neurogenic niches in our model, this would go beyond the scope of this review. Nevertheless, we included two new paragraphs on the neurogenic niches (lines 28-60), emphasizing the inter-species differences and differences in neurogenic rates.

We have removed the discussion regarding transplanting neuroblasts across different niches and revised the section on in vitro studies.

Round 2

Reviewer 2 Report

The authors have clarified most of the previous problems and increased the quality of this review.